# Remote Sensing of Snow Cover Variability and Its Influence on the Runoff of Sápmi's Rivers

Sebastian Rößler [1,*], Marius S. Witt [2], Jaakko Ikonen [3], Ian A. Brown [4]  and Andreas J. Dietz [1]

1 German Remote Sensing Data Center (DFD), German Aerospace Center (DLR) Muenchener Strasse 20, D-82234 Wessling, Germany; Andreas.Dietz@dlr.de
2 Department of Remote Sensing, Institute of Geography and Geology, University of Wuerzburg, Am Hubland, D-97074 Wuerzburg, Germany; Marius.Witt@stud-mail.uni-wuerzburg.de
3 Arctic Space Center, Finnish Meteorological Institute (FMI), Erik Palmenin Aukio 1, FI-00560 Helsinki, Finland; Jaakko.Ikonen@fmi.fi
4 Department of Physical Geography, Stockholm University, Svante Arrheniusväg 8, SE-10691 Stockholm, Sweden; Ian.Brown@natgeo.su.se
* Correspondence: Sebastian.Roessler@dlr.de; Tel.: +49-(0)-8153-28-3156

**Abstract:** The boreal winter 2019/2020 was very irregular in Europe. While there was very little snow in Central Europe, the opposite was the case in northern Fenno-Scandia, particularly in the Arctic. The snow cover was more persistent here and its rapid melting led to flooding in many places. Since the last severe spring floods occurred in the region in 2018, this raises the question of whether more frequent occurrences can be expected in the future. To assess the variability of snowmelt related flooding we used snow cover maps (derived from the DLR's Global SnowPack MODIS snow product) and freely available data on runoff, precipitation, and air temperature in eight unregulated river catchment areas. A trend analysis (Mann-Kendall test) was carried out to assess the development of the parameters, and the interdependencies of the parameters were examined with a correlation analysis. Finally, a simple snowmelt runoff model was tested for its applicability to this region. We noticed an extraordinary variability in the duration of snow cover. If this extends well into spring, rapid air temperature increases leads to enhanced thawing. According to the last flood years 2005, 2010, 2018, and 2020, we were able to differentiate between four synoptic flood types based on their special hydrometeorological and snow situation and simulate them with the snowmelt runoff model (SRM).

**Keywords:** remote sensing; snow parameters; snow variability; MODIS; snow hydrology; spring flood; Sápmi; Mann-Kendall test; snowmelt runoff model

## 1. Introduction

The hydroclimatic regime in the Arctic and sub-Arctic is predominantly nival and spring floods triggered by snowmelt occur every year [1–4]. Spring floods are defined as the characteristic high discharge events that occur after prolonged low flows over the winter. Characteristics of spring floods are discharge peaks demonstrating rapid acceleration in discharge followed by distinct kurtosis; these indicate the sudden onset of widespread snowmelt in spring [5]. As a major component of the cryosphere, the seasonal snow cover and its alteration is a clearly visible indication of changing climatic conditions.

Climate change has and will continue to result in two effects: shorter winter conditions and more inter-annual variability. The increasing variability of snow in mountains for example has become a serious problem with regular seasons experiencing low snow accumulation. Ski areas in both the Alps [6,7] and the Pyrenees [8] will have to cope with unpredictable snow conditions. In addition, although most climate models predict a decrease in snow cover in the northern hemisphere [9,10], contrary developments have also been observed [11,12]. The spatial complexity of snowfall makes prediction and

generalization difficult. That said, an increase in moisture availability is expected to lead to more snow in the Scandinavian highlands and less snow, more rain in the lowlands and coastal regions [13]. This will have implications for the timing and intensity of the spring flood and for the generation of floods associated with extreme weather events. Currently snowmelt is the main mechanism of flood generation in northern Fenno-Scandia [14]. The winter of 2019/2020 was particularly snowy in large parts of Scandinavia—especially in the north. This led to severe spring floods in the Sápmi region, particularly in the Kemijoki river system in Finland [15]. In the unregulated rivers of northern Sweden, the spring floods were less pronounced in 2020 (compared to the last major floods 2010 and 2018), which raises the question of the driving forces.

In this investigation we analyze the spatio-temporal patterns in snow cover and runoff in the Sápmi region for the period 2001–2020 and the factors affecting discharge extremes. This period exhibits high variability in the timing of the onset, longevity, and end of the snow season. Since snow deposits are the water reservoirs for nival rivers in winter, our main focus is on analyzing the annual snow cover: remote sensing methods are suitable for this task [16]. Statistical analysis of upland catchment data shows that discharge is most often correlated with temperature but not precipitation, even accounting for lags. As a proxy for the water reservoir stored in the snowpack we use snow-covered area estimates from MODerate resolution Imaging Spectroradiometer (MODIS) satellite data. The snow data derived from MODIS has a long history of studying the seasonal changes in snow cover, and has applications in the Himalayas [17,18], in East Asia [19,20], Central Asia [21–23], North America [24,25], North Africa [26,27], and in Europe [28–30]. However, the use of optical data to determine snow cover requires methods for interpolation in the case of cloud cover and—at higher latitudes—missing data due to the polar night. To solve this problem, the DLR's Global SnowPack processor was used in this study [31]. To study the effects of the extraordinary snow conditions in northern Fenno-Scandia, rivers were chosen that were not regulated by dams for flood protection or hydropower generation. In northern Sweden, the rivers Torneälven, Kalixälven, Piteälven, and Vindelälven (a tributary of the Umeälven) were chosen. As national rivers, they are not influenced by hydropower exploitation. In Finnish Lapland, the Kemijoki is the largest river alongside the unregulated border river with Sweden (the Muoinoälven, which flows into the Torneälven). Most of the Kemijoki river is heavily regulated, so only the Ounasjoki tributary (which flows into the Kemijoki in Rovaniemi) and the Kemihaara (upper reaches of the Kemijoki watershed in the northwestern catchment) area are selected.

In this study, we will examine the dynamic of snow cover and the associated effects on runoff over the last 20 years for the selected catchments. The variability of the satellite-derived snow cover duration (SCD) will be analyzed and compared to the runoff data with a special focus on hydrological events (floods and droughts) for each catchment. The first goal is therefore to analyze whether these events can be explained by the variability of the snow cover alone. In addition, freely available meteorological data from several weather stations on air temperature and precipitation are included in the study. From these datasets, various robust time measures [1,32,33] were extracted and their relationships to one another were examined using correlation analysis. Furthermore, we investigated their temporal development using trend analysis in the observation period between the hydrological years 2001 and 2020. The second goal is to find out which interdependencies exist between the examined parameters and whether an increase or decrease in extreme hydrological events is to be expected (in particular the maximum discharge of the spring flood and its timing). In addition to the snow cover and topographical information derived from a digital elevation model, the meteorological information is included in the simple Snowmelt Runoff Model (SRM) [34]. The third aim of the study is to find out whether the occurrence of extreme hydrological events can be determined using this simple model.

The first research question is as follows: Is satellite-derived snow cover area alone already suitable to describe the occurrence of hydrological events? The second research question is as follows: Are there trends in the occurrence of hydrological extreme events

within the period under consideration and can these be traced back to extraordinary meteorological events? The third research question is as follows: Can the hydrological events be adequately mapped using the simple SRM? We address these questions throughout the paper within the following sections. In Section 2 the material and methods are presented including a description of the studied catchments. Section 3 contains the results of the snow cover and runoff development, the meteorological factors as well as their trends and interdependencies. Section 4 serves to synthesize the results and define various snow conditions that lead to floods. In Section 5, a conclusion is drawn and an outlook on subsequent challenges is given.

## 2. Materials and Methods

### 2.1. Snow Cover

Snow coverage in this study is derived from the MODerate resolution Imaging Spectrometer (MODIS). The daily snow cover product MOD10A1 (Terra) and MYD10A1 (Aqua) from the version 6 [35] are derived from the normalized difference snow index (NDSI) and made available free of charge by the National Snow & Ice Data Center (https://nsidc.org/data/MOD10A1, https://nsidc.org/data/MYD10A1, (accessed on 10 October 2020). Since operational data from MODIS Terra are available since 2000-02-24 and from MODIS Aqua since 2002-07-04, we examined the last 20 hydrological years (2001 to 2020). We define the hydrological year in the northern hemisphere beginning with 1 September of the previous year and ending on 31 August of the current year. This results in a total of 7305 MODIS Terra scenes and 6634 MODIS Aqua scenes. Each of the square MOD10A1/MYD10A1 tiles has a side length of 1200 km (this corresponds to approx. 10°) and is in the equal-area sinusoidal projection [35]. Our investigation area is covered by the tiles h18v02 and h19v02. The adverse effects of clouds and the polar night in these high latitudes make effective interpolation of the data essential (Figure 1).

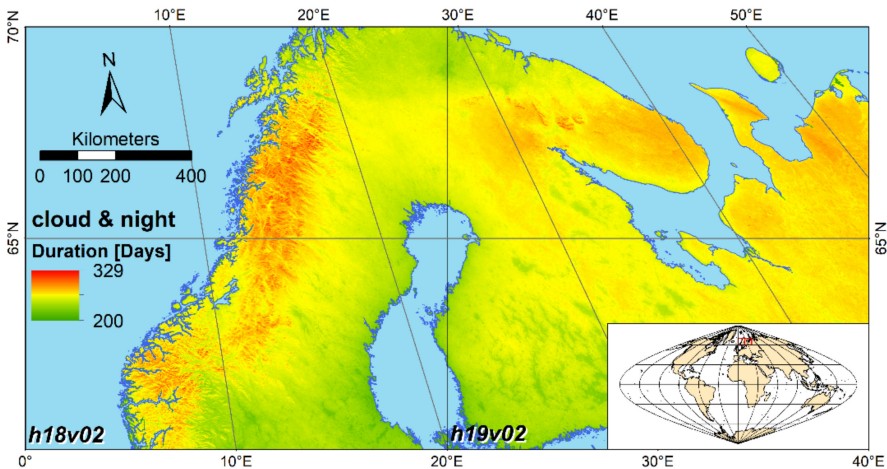

**Figure 1.** Number of days that are either influenced by clouds and polar night (arithmetic mean of the hydrological years 2001 to 2020).

The DLR's Global SnowPack (GSP) processor [31] was used to achieve cloud-free scenes. In the course of the interpolation, all "no-data" pixels are treated as "clouds" for the sake of simplicity. Missing information, e.g., due to polar night, is treated in the same way as persistent cloud cover.

First there is a binary decision between "snow" and "non-snow" from the layer "NDSI_snow_cover" of the HDF-file. Since version 6 of the daily MODIS snow product now provides "raw" NDSI values, a threshold needs to be defined to classify pixels into "snow" and "no snow" [36]. Each pixel that may contain snow has an NDSI value between 0 and 100. Since freshly fallen snow has a much higher reflection than old snow, the threshold value for detection must also be adjusted over the course of the year [37]. We

also differentiate between "snow in forests" ($NDSI_{sif}$), which has a lower NDSI due to the parts of the plants shown, and "snow in non-forests" ($NDSI_{sinf}$). The thresholds are calculated using Equations (1) and (2):

$$NDSI_{sif} = 12.673 \times \cos\left[\frac{2\pi}{365} \times (190 - DOHY)\right] + 39.471 \tag{1}$$

$$NDSI_{sinf} = 12.780 \times \cos\left[\frac{2\pi}{365} \times (226 - DOHY)\right] + 51.635 \tag{2}$$

where DOHY is the day of the hydrological year. These thresholds have been empirically defined by using the MODIS tiles h18v04, h18v03 and h18v04 during the hydrological years 2010 to 2014. The snow-covered area has been validated against Landsat derived snow coverage in order to determine the best fitting threshold without underestimating or overestimating the actual snow coverage.

In a first step, the daily observations of Terra and Aqua are combined to already eliminate cloudy pixels. However, due to a malfunction of the 1.6 µm band, the MODIS sensor on Aqua has a less precise cloud mask [38], so that the data from MODIS Terra is given priority and serves as a basis. All pixels classified as clouds are now checked in the Aqua data to determine whether they are not covered by clouds and, in this case, adopted. After that the remaining cloudy pixels are interpolated considering the previous day and the following day. If the degree of cloud cover is below a specified threshold value (in our case 30%), a topographical interpolation is then carried out. For this purpose, we use a digital elevation model (DEM), specifically the Global Multi-resolution Terrain Elevation Data 2010 (GMTED2010, available on Earth Explorer: https://earthexplorer.usgs.gov/, accessed on 30 January 2020) [39] that was interpolated and projected to the MODIS tiles. The maximum elevation of "snow-free" pixels as well as the minimum elevation of "snow-covered" pixels is determined. If the elevation of a cloud pixel lies below the minimum height of "snow-covered" pixels, this pixel is determined to be "snow-free"; conversely, cloud pixels that lie above the maximum elevation of "snow-free" pixels are classified as "snow-covered". As a last step, the remaining pixels are interpolated by iterating over the entire data stack with increasing temporal distance until there are no more cloud pixels left.

From the resulting cloud-free daily snow cover maps, information for selected basins was derived by clipping the raster using a shapefile outlining the basin area. From these extracted raster, the snow cover duration ($SCD$) was calculated with Equation (3) [29]:

$$SCD = \sum_{i=1}^{n}(s_i) \tag{3}$$

where $n$ is the day of the hydrological year, and $s_i$ is the daily cloud-free snow raster (pixels were recoded to 1 for "snow" and 0 for "snow free"). Since we are especially interested in the distribution between early season SCD ($SCD_{ES}$), and late season SCD ($SCD_{LS}$) the snow cover duration was divided into these two parts using Equations (4) and (5):

$$SCD_{ES} = Fd - SCD_{bFd} \tag{4}$$

$$SCD_{LS} = Fd + SCD_{aFd} \tag{5}$$

where $Fd$ is a fixed date (we select 15 January due to maximum snow coverage), $SCD_{bFd}$ and $SCD_{aFd}$ is the snow cover duration before and after this fixed date. To eliminate the lakes in the snow map, a water mask was created for each year and all pixels were masked that were recognized as water areas in at least 5% (18 days) of the cases.

Although the MODIS 8-day snow product (M*D10A2) is of excellent quality and is used in many studies, we use the daily product (M*D10A1) due to two reasons: The 8-day product only contains a cloud mask for the entire eight days (which is always cloud-covered within the period), for the individual days the snow mask is only available in binary form

as "snow/non-snow", information on why pixels are classified as "non-snow" does not exist for the individual days. This is unfavorable for our purpose, which aims for daily snow cover, as it is not known which pixels are actually clouds and should be interpolated. Another reason is the binary classification that has already taken place, which means that our NDSI threshold-based determination (temporal variable and including land cover types) can no longer be carried out.

### 2.2. Hydrology

Hydrological information (i.e., discharge data) has been obtained for the selected river systems shown in Figure 2. For the four river systems that are (largely) on Swedish territory, discharge data were obtained from the Swedish Meteorological and Hydrological Institute (SMHI) (https://vattenwebb.smhi.se/station/, accessed on 5 October 2020). Information from the finish river system was obtained from the Finnish Environment Institute (SYKE, https://www.syke.fi, accessed on 3 October 2020).

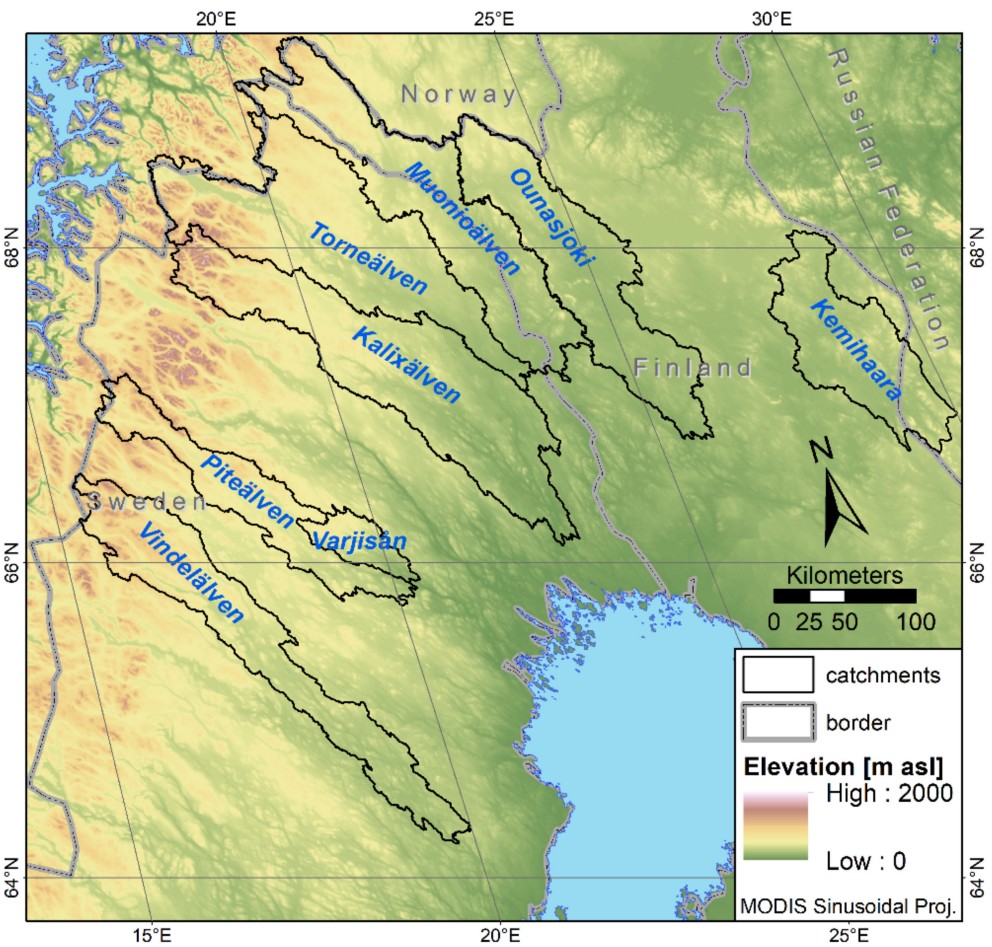

**Figure 2.** Main study area of Sápmi showing the location of the eight selected catchments.

In total there are discharge measurements from 36 stations available. However, since we only consider the largest of the overlapping catchment areas and have defined a minimum size of 1000 km$^2$ (due to the spatial resolution of MODIS), eight catchment areas were selected for further analysis. An overview about the catchment properties is given in Table 1. The shapefiles of these sub basins are used to extract spatial statistics from snow coverage elevation.

**Table 1.** Catchment properties of the eight selected sub basins. The Basin ID is the official number of the catchment area according to SMHI (Sweden) and SYKE (Finland).

| River | Basin ID | Country | Area [km²] | Elevation [m MSL] | | |
| --- | --- | --- | --- | --- | --- | --- |
| | | | | Min | Max | Mean |
| Kalixälven | 0017 | SE | 23,102.9 | 41 | 1962 | 406.1 |
| Piteälven | 1387 | SE | 6930.9 | 129 | 1634 | 593.8 |
| Varjisån | 1706 | SE | 1908.4 | 78 | 679 | 396.0 |
| Torneälven | 2012 | SE | 11,038.1 | 154 | 1828 | 536.4 |
| Vindelälven | 2237 | SE | 11,846.4 | 162 | 1531 | 515.0 |
| Muonioälven | 2395 | SE/FI | 14,477.1 | 140 | 1471 | 412.6 |
| Kemihaara | 6501700 | FI | 8464.2 | 157 | 617 | 270.8 |
| Ounasjoki | 6503600 | FI | 11,660.0 | 102 | 746 | 270.7 |

Figure 3 shows the elevation distributions (hypsometric curves) of all selected basins. While the rivers Ounasjoki, Kemihaara, and Varjisån are almost exclusively below 500 m, the catchment areas of the others have a pronounced topographical gradient.

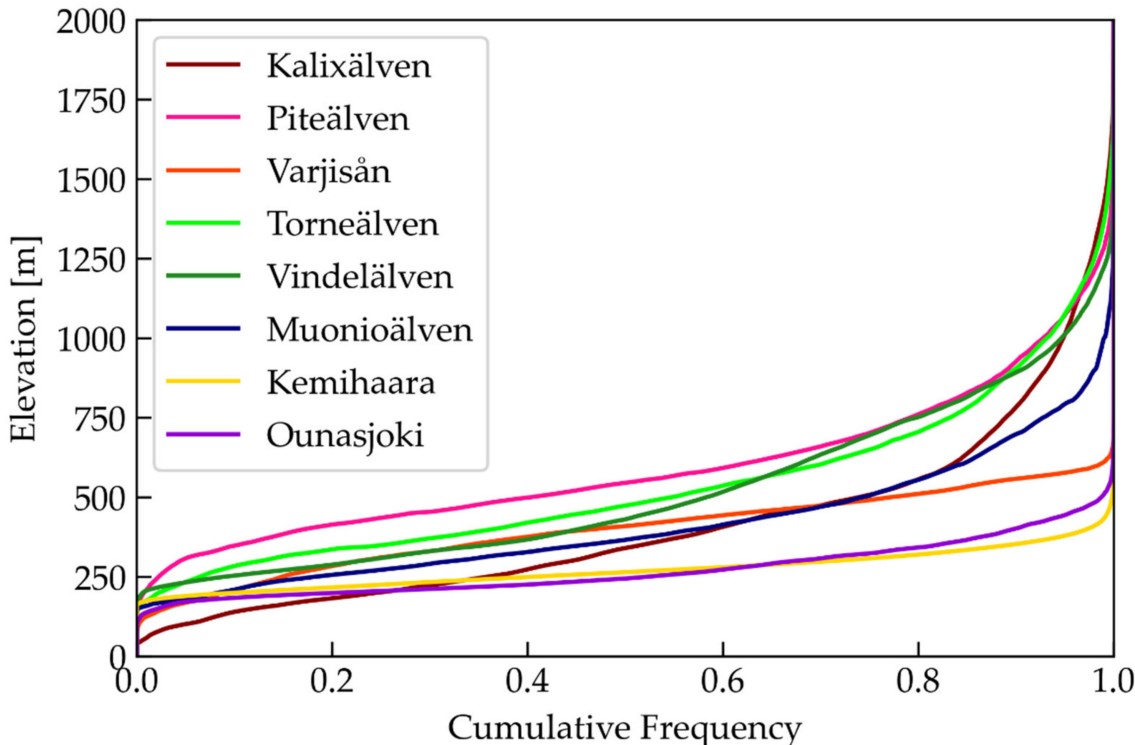

**Figure 3.** Hypsometric curves of the eight selected catchments.

In terms of runoff data, we are particularly interested in the spring runoff (start and end of snowmelt). To detect timing changes in the runoff behavior [32], we extract the maximum discharge (both intensity and time) for each hydrological year. The use of fixed runoff quantiles during spring runoff has been proven as a suitable tool to detect changes [33,40]. For our application we use the period between 1 April to 30 June, as snowmelt can occur as early as April. We calculate a selection of cumulative runoff intervals: Q5% as the point in time at which 5% of the cumulative runoff of the observed period occurred [1]. The times Q10%, Q50%, Q90%, and Q95% are calculated similarly.

*2.3. Meteorology*

Daily data on air temperature (in the following always the air temperature measured at a height of 2 m) and precipitation were downloaded for Finland from the Finnish Meteorological Institute (FMI; https://en.ilmatieteenlaitos.fi/download-observations, accessed on 11 October 2020) and for Sweden from the Swedish Meteorological and Hydrological Institute (SMHI; http://www.smhi.se/data/meteorologi/, accessed on 9 October 2020). The station data had to contain both air temperature and precipitation data and should also be as complete as possible for the investigated period. A total of 44 stations met these criteria. Four meteorological stations should be available for each sub basin. To ensure this, the closest stations were searched for with a buffer of up to 50 km around the respective catchment areas. This resulted in 27 stations that were used for further analyses. The mean values for air temperature and precipitation of the individual stations for each day were calculated for each selected sub basin from the station data. The air temperatures were previously converted to the mean terrain height of the basin before this was included in the calculation. The used vertical temperature decrease of 0.65 K per 100 m corresponds to the tropospheric temperature gradient of the international standard atmosphere (ISA) established by the ICAO (International Civil Aviation Organization).

To determine the beginning and the end of the snow cover, the air temperatures were summed over the hydrological year. The maximum of the resulting curve within the autumn and winter months (September—February) was selected as the snow cover start (SCS), and the minimum of the remaining months (March—August) as the snow cover melt (SCM).

Wind has also a decisive influence on the redistribution and melting of snow and, depending on the exposure and relief, large differences can occur on a small scale. However, since the meteorological data in this study are mainly used as input parameters for the snowmelt runoff model, which does not take the wind into account, it is accordingly neglected.

*2.4. Statistical Analysis*

As mentioned in the introduction, with regard to the temporal development of the snow and the associated runoff, the relationships and dependencies of the parameters are of interest to one another and the temporal development of the parameters in the observation period in order to be able to derive trends if necessary. To identify relationships and controlling factors of the 14 timing measures (Table 2) used within this study, we calculated the Pearson's correlation coefficient for each pair of hydroclimatic variables [1]. A trend analysis for each measure (and for each basin) was performed using the non-parametric Mann-Kendall (MK) trend test [41,42]. This test has been proven as a valuable tool to detect changes in snow determined changes of runoff [1,18,33,43,44].

**Table 2.** Measures for spring flood timing and intensity.

| Abbreviation | Unit | Description | Source |
|---|---|---|---|
| Year | Year | Hydrological Year | - |
| $SCD_{ES}$ | Days | Early Season Snow Cover Duration | Global SnowPack |
| $SCD_{LS}$ | Days | Late Season Snow Cover Duration | Global SnowPack |
| SCS | DOHY [1] | Snow cover start derived from daily mean air temperature | Meteorology |
| SCM | DOHY | Snow cover melt derived from daily mean air temperature | Meteorology |
| Snow_Prec | mm | Cum. sum of (solid) precipitation between SCS and SCM | Meteorology |
| Spring_Rain | mm | Cum. sum of precipitation between SCM and 30 June | Meteorology |
| Qmax | $m^3/s$ | Peak discharge | Hydrology |
| DOHY_Qmax | DOHY | Time of peak discharge | Hydrology |
| Q5% | DOHY | Time where 5% of the cum. discharge occur | Hydrology |
| Q10% | DOHY | Time where 10% of the cum. discharge occur | Hydrology |
| Q50% | DOHY | Time where 50% of the cum. discharge occur | Hydrology |
| Q90% | DOHY | Time where 90% of the cum. discharge occur | Hydrology |
| Q95% | DOHY | Time where 95% of the cum. discharge occur | Hydrology |

[1] Abbreviation for "day of hydrological year".

### 2.5. Snowmelt Runoff Model

The Snowmelt Runoff Model (SRM) was developed in 1975 by Martinec [45] and can be used to model daily discharge in mountainous areas. It has been successfully applied to basins ranging from 0.76 to 917,444 km$^2$ in size, and with elevation ranges of up to 8840 m. The daily discharge is calculated using Equation (6) [34]:

$$Q_{n+1} = [c_{Sn}a_n(T_n + \Delta T_n)S_n + c_{Rn}P_n]\frac{A}{8.64}(1 - k_{n+1}) + Q_nk_{n+1}, \qquad (6)$$

where $Q$ is the daily discharge (m$^3$/s), $c_S$ and $c_R$ are the runoff coefficients expressing the ratio of loss between runoff and precipitation for snow and rain. $a$ is the degree-day factor ($\frac{cm}{°C·d}$)—the snowmelt depth from one degree-day. $T$ is the number of degree-days (°C·d). $\Delta T$ is the air temperature correction, calculated with the height difference between measurement height and mean hypsometric basin height, using an air temperature lapse rate $T_{lapse}$ of 0.65 $K/100m$. $S_n$ is the percentage of snow cover in the basin area. $P$ is the measured precipitation in cm. The model expects a critical air temperature $T_{crit}$: If $T < T_{crit}$, the precipitation will be stored as snow and melts when melting conditions occur. $A$ is the basin area in km$^2$. $k$ is the recession coefficient, governing the discharge in periods without rainfall or snow melt. For basins with elevation ranges exceeding 500 m, the basin will be subdivided into zones of 500 m each. In that case, the discharge is calculated separately for each zone and summed, before multiplying with $1 - k_{n+1}$. To evaluate the model output, we use the Nash-Sutcliffe efficiency coefficient $E$ [46], calculated by Equation (7):

$$E = 1 - \frac{\sum_{i=1}^{n}(Q_i - Q'_i)^2}{\sum_{i=1}^{n}(Q_i - Q_m)^2}, \qquad (7)$$

where $Q_i$ is the measured discharge, $Q'_i$ is the simulated discharge and $Q_m$ the average discharge for the simulation period. The recession parameter $k$ can be derived from historic discharge data with $k = Q_{m+1}/Q_m$ with $Q_m$ being the discharge during a true recession flow period without rainfall or snowmelt. Since the Parameters $c_R$, $c_S$ and $a$ must have a value greater than 0 and less than 1, they were evaluated in the range from 0.05 to 0.99 (using steps of 0.02) to find the combination with the highest Nash-Sutcliffe efficiency as described in [47].

## 3. Results

### 3.1. Snow Cover

Figure 4 shows the mean snow cover duration (SCD) for the hydrological years 2001 to 2020. Snow cover is most persistent in the Scandinavian Mountains, Sápmi, and the Kola Peninsula. The lakes are also easily recognizable due to their relatively small number of snow-covered days. This is due to the fact that they still have an open water surface next to snow-covered land, or they are covered with lake ice (without snow cover), which forms a separate class in the MODIS data.

Figure 5 shows the deviation of the hydrological year 2020 compared to the long-term mean (2001–2019). In most parts of the region, SCD was up to 50 days longer compared to the mean duration. A closer look at the Early Season SCD the (SCD$_{ES}$) and Late Season (SCD$_{LS}$) revealed that snow coverage both started earlier and lasted longer this year. The pattern is particularly marked in the densely forested lower Umeälven and Piteälven catchments and other heavily forested locations in northern Fenno-Scandia.

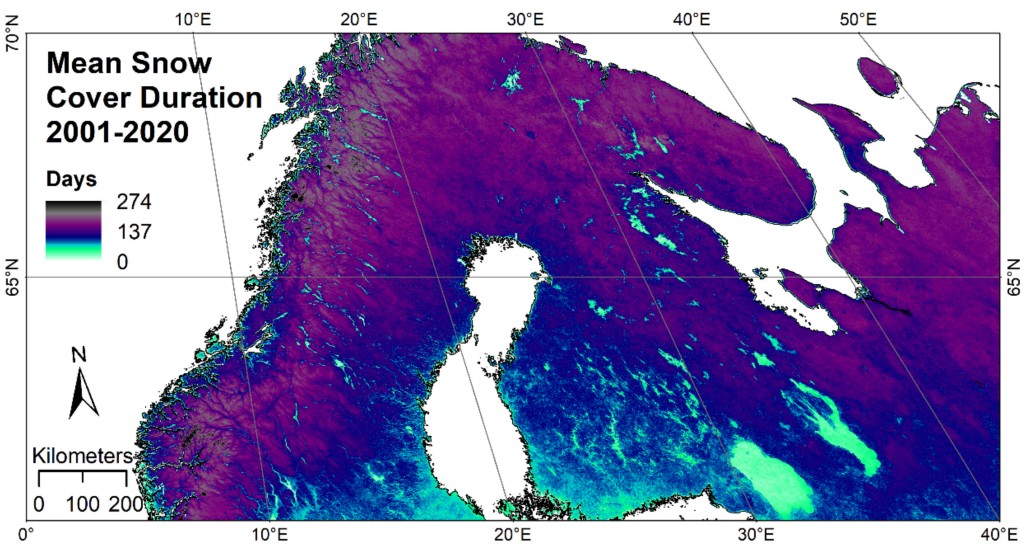

**Figure 4.** Arithmetic mean of the yearly snow cover duration for the hydrological years 2001 to 2020 for the tiles h18v02 and h19v02.

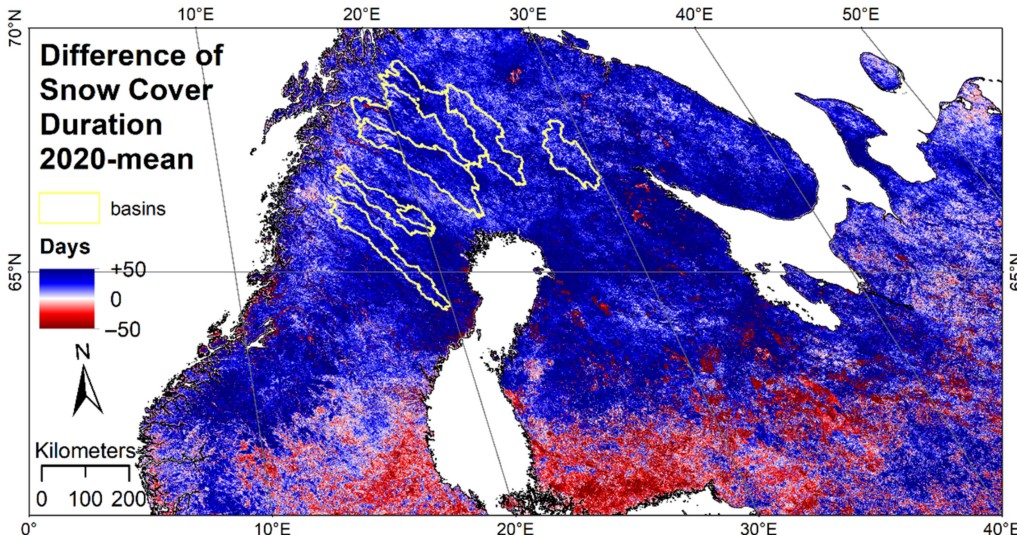

**Figure 5.** Deviation of snow cover duration for the hydrological year 2020 compared to the mean SCD calculated from the hydrologic year 2001 to 2019.

Figure 6 shows the individual snow coverages (weekly mean) for each catchment area. Additionally, $SCD_{ES}$ (solid line) and $SCD_{LS}$ (dashed line), both calculated according to Equations (4) and (5), are shown. What is initially noticeable is the long range between November and January in which values stagnate—this is caused by the polar night (hashed areas in Figure 6). This is the time when the greatest uncertainties regarding snow cover prevail, as the data are interpolated over very long periods of time. Since there is still no snow cover, especially in early winter, this underestimation will be continued until data are available after the polar night. The year 2012 shows very little snow cover in all catchment areas until the end of the polar night, the snow accumulation started extremely late this year. Also, years with very long snow cover are recognizable: The hydrological year 2020 shows an extraordinarily long snow cover in all catchment areas. The year 2010 was a little less pronounced, but still exceptionally snowy. However, the floods of 2018 did not show any extremes in terms of snow cover, the causes here are probably due to hydrometeorological factors.

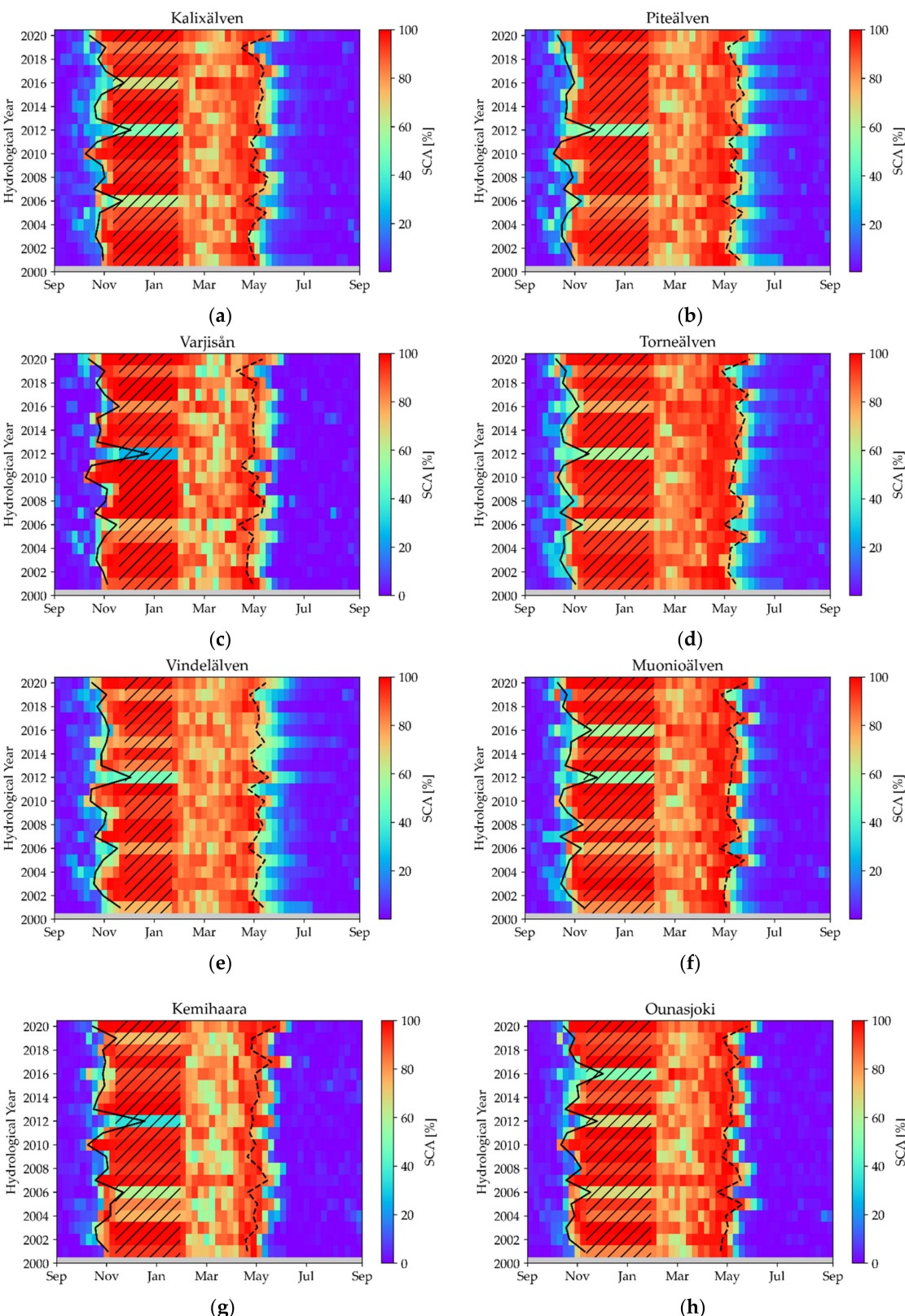

**Figure 6.** Weekly mean of snow coverage for the hydrologic years 2001 to 2020 for (**a**) Kalixälven, (**b**) Piteälven, (**c**) Varjisån, (**d**) Torneälven, (**e**) Vindelälven, (**f**) Muonioälven, (**g**) Kemihaara and (**h**) Ounasjoki. The hatched area shows the polar night (depending on the catchment area between weeks 10 and 22), the solid line shows the calculated $SCD_{ES}$, the dashed line the time of the $SCD_{LS}$.

### 3.2. Hydrology

The discharge data for the hydrological years 2001 to 2020 showed a very heterogeneous picture for the relatively narrow geographic area. However, all rivers have a pronounced nival runoff regime with an annual runoff peak in spring. We identified four years with an extremely high runoff: the years 2005, 2010, 2018 and 2020. The following Figures 7–9 show the discharge behavior as a color-coded grid (x-axis: day of hydrological year; y-axis: hydrological year) for these flood years. The absolute discharge maxima are each marked with a "∧"-sign, the minimum peak discharge with a "∨"-sign.

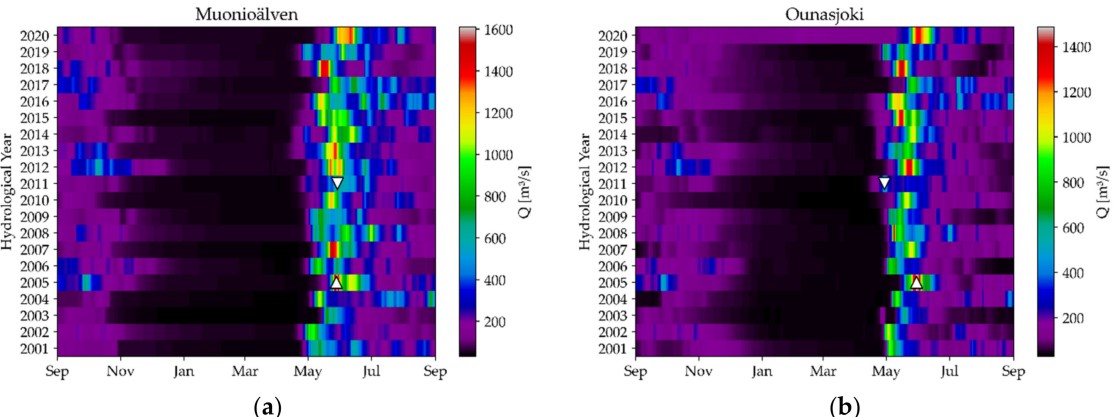

**Figure 7.** Discharge plots of (**a**) Muonioälven and (**b**) Ounasjoki. The upward pointing triangle indicates the maximum spring flood, the downward pointing the minimum.

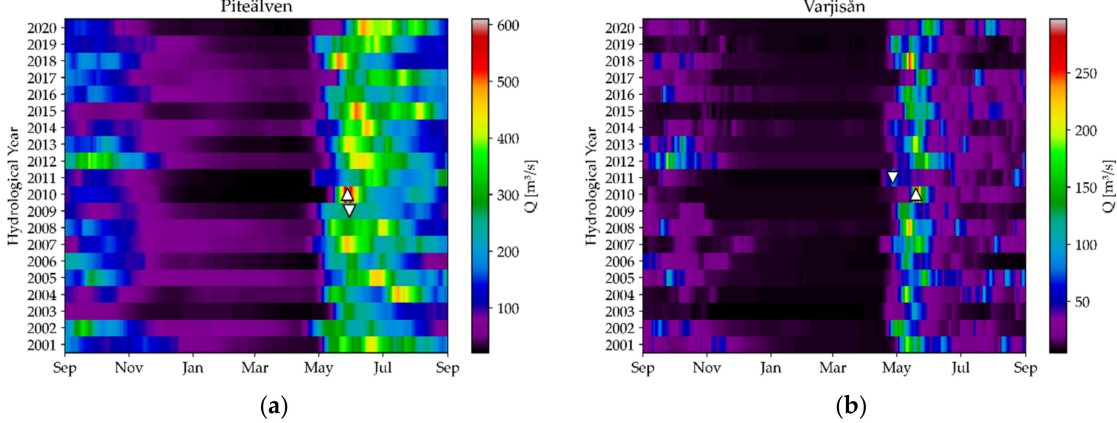

**Figure 8.** Discharge plots of (**a**) Piteälven and (**b**) Varjisån. The upward pointing triangle indicates the maximum spring flood, the downward pointing the minimum.

The Swedish-Finnish border river Muonioälven (Figure 7a) carried 1610 m$^3$/s on 28 May 2005, which corresponds to about nine times the mean discharge (178 m$^3$/s) over the period under consideration. In the same year, the neighboring Finnish river Ounasjoki (Figure 7b) reached its peak on 30 May 2005 with a discharge of 1486 m$^3$/s (average 142 m$^3$/s).

In 2010, the Swedish rivers Piteälven (Figure 8a) reached 610 m$^3$/s on 28 May 2010 (mean: 127 m$^3$/s) and its northern tributary Varjisån (Figure 8b) with 297 m$^3$/s on 19 May 2010 (mean: 24 m$^3$/s) their highest values. This year was also an important flood year for other rivers: the Vindelälven reached its second highest discharge, Kalixälven and Kemijoki their third highest.

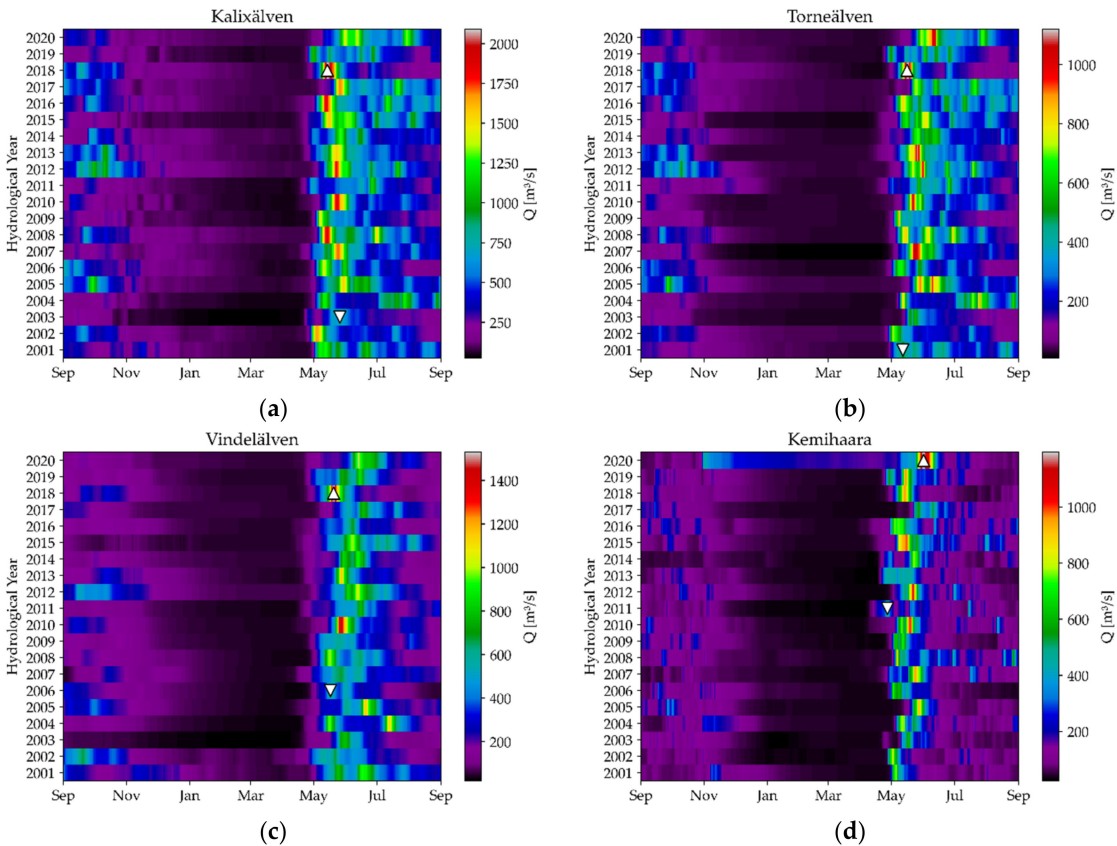

**Figure 9.** Discharge plots of (**a**) Kalixälven, (**b**) Torneälven, (**c**) Vindelälven, and (**d**) Kemihaara. The upward pointing triangle indicates the maximum spring flood, the downward pointing the minimum.

The year 2018 was the most pronounced flood year in the period under consideration: the rivers Kalixälven (2090 m$^3$/s on 14 May 2018; mean 314 m$^3$/s; (Figure 9a), Torneälven (1120 m$^3$/s on 16 May 2018; mean 148 m$^3$/s; (Figure 9b) and Vindelälven (1530 m$^3$/s on 20 May 2018; mean 186 m$^3$/s; (Figure 9c) reached their highest discharge. The rivers Piteälven and Varjisån reached their second highest discharge, the rivers Muonioälven and Ounasjoki their third highest. In 2020, only the Kemijoki (Figure 9d) reached its highest discharge with 1196 m$^3$/s on 31 May 2020 (mean 120 m$^3$/s). For the Kemijoki, however, the high base runoff in winter 2020 is exceptional.

The hydrological data for the start of the hydrological year also show interesting patterns. In all catchments except Kemihaara 2012 exhibits elevated runoff in September and October. The start of the hydrological year also shows higher than average discharge in 2005 in all catchments, especially in the Kalixälven and Torneälven. Early season discharge is found to be more common in the Piteälven and Kalixälven.

### 3.3. Meteorology

Figures 10 and 11 show the weekly mean air temperatures and the summed weekly precipitation totals for the hydrological years 2001 to 2020. The solid vertical lines show the SCS and SCM derived from air temperature, the dashed lines SCD$_{ES}$ and SCD$_{LS}$ from GSP, respectively. At Kalixälven (Figure 10a), for example, the lowest runoff peak in 2003 was due to the exceptionally low snow precipitation. In 2016, there was an extraordinary amount of snow precipitation, but the air temperature development in spring was more moderate, so that no record floods occurred. For Piteälven (Figure 10b) and Varjisån (Figure 10c), the highest runoff occurred in 2010, where the air temperatures were relatively low during winter enabling a large accumulation of snow. The years 2003 and 2006 were the years with the lowest snow precipitation and consequently little runoff. For Torneälven

(Figure 10d) the lowest runoff peak occurred in 2001, the precipitation was average this year, snow accumulation started late and ended early. The year 2018 (The largest spring runoff of the Torneälven) was not exceptional in terms of precipitation; however, there was a sharp rise in weekly mean temperature at the end of the snow season.

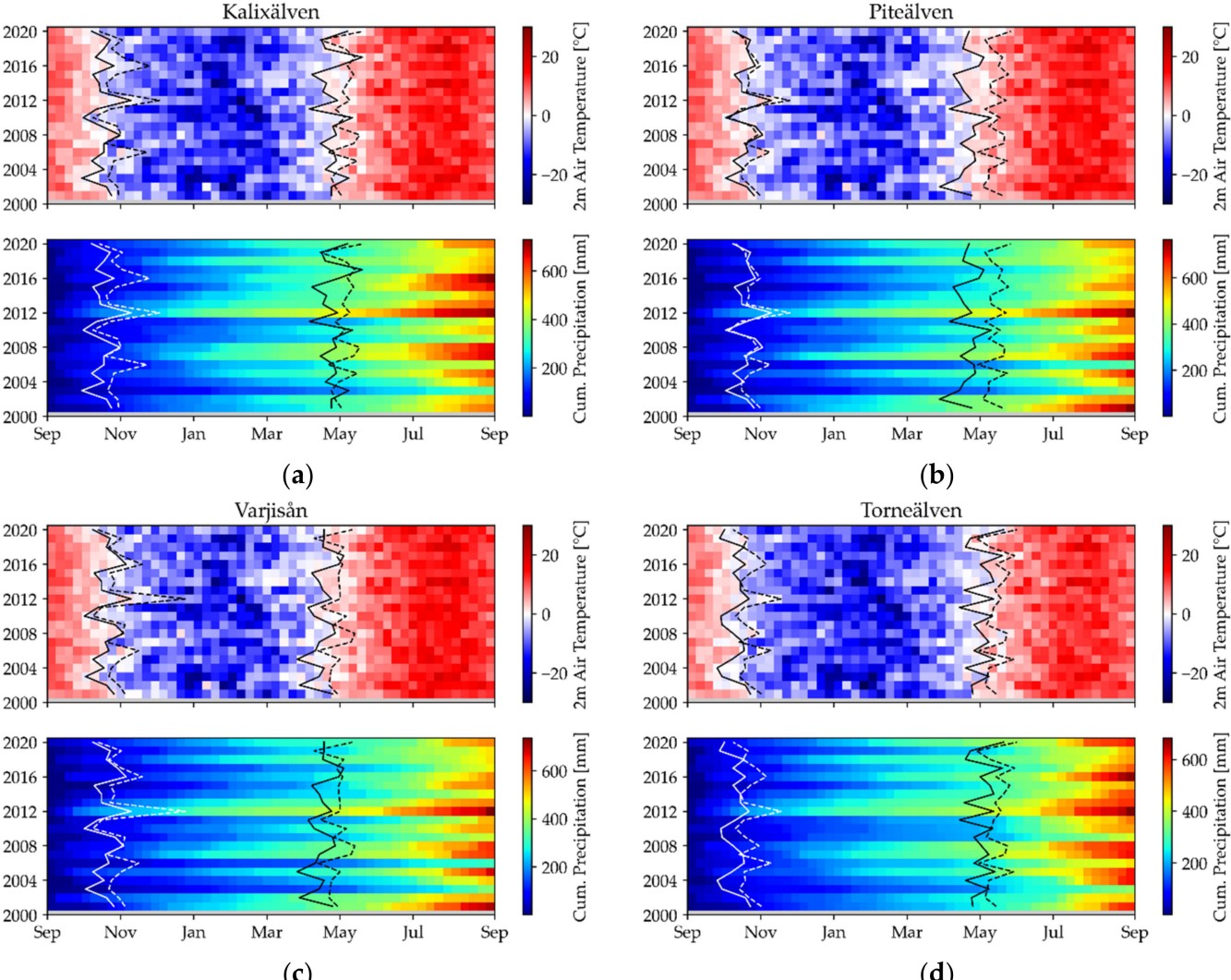

**Figure 10.** Weekly mean air temperature and weekly sum of precipitation for (**a**) Kalixälven, (**b**) Torneälven, (**c**) Vindelälven, and (**d**) Kemihaara. The earlier lines show the snow cover start, the later the beginning of snowmelt. Solid lines are derived from the satellite product; dashed are derived from the air temperature.

At the Vindelälven (Figure 11a), the lowest discharge peak occurred in 2006 where the winter precipitation was low. The record runoff in 2018 is again not exceptional in terms of precipitation amount. However, the temperature throughout the winter was low so that snow could accumulate well. At the end of the snow cover season, the average temperature rose sharply. At Mounioälven (Figure 11b), the minimal discharge peak was in 2011, where the amount of winter precipitation was low. In 2005 (the highest peak) the amount of winter precipitation was high. The Kemihaara (Figure 11c) received a large amount of winter precipitation in 2020 (highest peak) and the snow cover extended well into spring. In 2011 (lowest peak), the amount of winter precipitation was low. For Ounasjoki (Figure 11d), this was also the case in 2011 (lowest peak). The highest discharge occurred in 2005, where a large amount of snow was received in late winter.

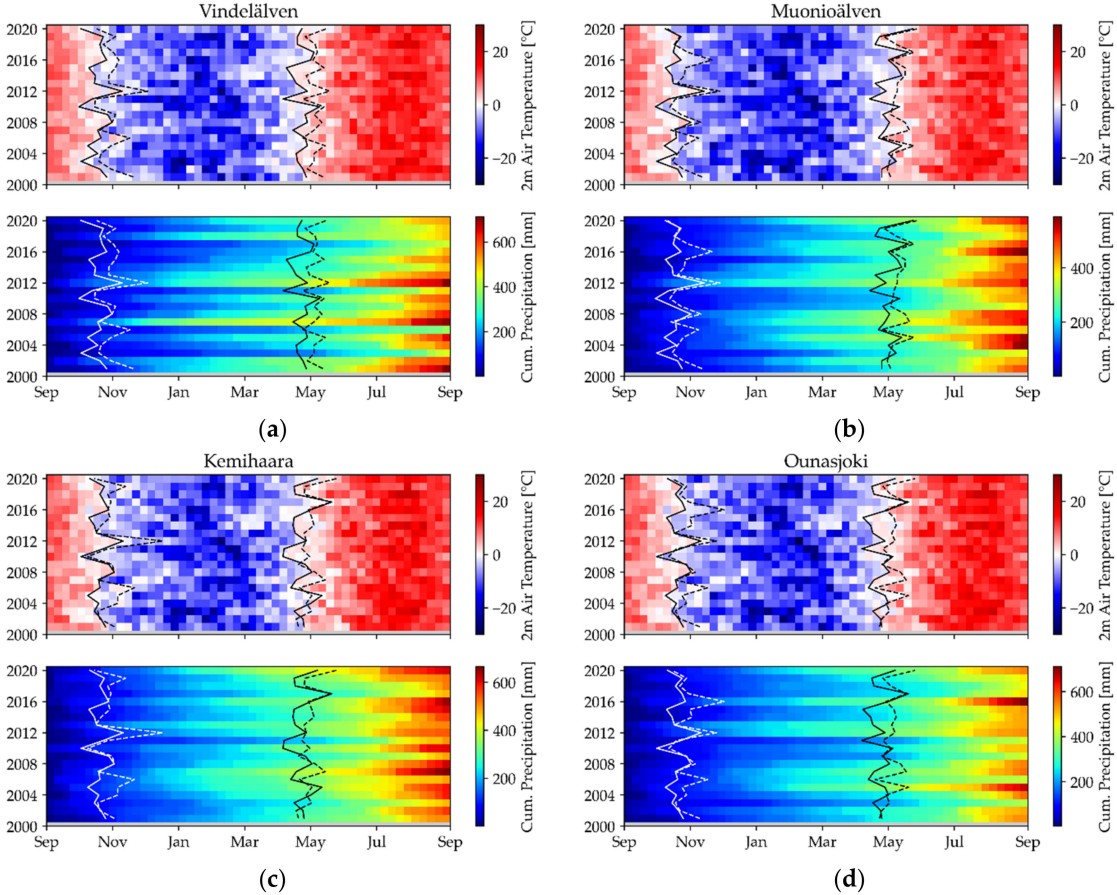

**Figure 11.** Weekly mean air temperature and weekly sum of precipitation for (**a**) Vindelälven, (**b**) Mounioälven, (**c**) Kemihaara, and (**d**) Ounasjoki. The earlier lines show the snow cover start, the later the beginning of snowmelt. Solid lines are derived from the satellite product; dashed are derived from the air temperature.

The early season meteorological data show high variability in precipitation but less in air temperature. The years 2015 and 2016 appear to have experienced elevated precipitation in September and October in several of the studied catchments. The years 2002 and 2012, years in which September and October discharge was higher than normal, are unremarkable in terms of air temperature but show increased precipitation. However, 2012 received also widespread early snowfall. This event, while less than 10 cm deep, was relatively short lived for large parts of the study area suggesting a snowmelt flood-event occurred despite mild air temperatures. This event can be recognized by the clear differences in the timing measures for the beginning of the snow in Figures 10 and 11. While the snow accumulation begins according to the air temperature threshold ("SCS"), it does not take place until some time later, according to the satellite-derived product ("SCD$_{ES}$"). This suggests that the snow cover only existed for a short time and thawed again.

### 3.4. Statistics

The following Figures 12 and 13 shows the correlation matrices for each catchment area. A correlation coefficient of 1 means a perfect linear relationship, the sign indicates whether this is positive or negative. If the value is above 0.8 there is a strong relationship, if it is above 0.6 the relationship is moderate, values above 0.4 are regarded as weak relationships. For all rivers there is a more or less strong connection between "SCD$_{ES}$" and "SCS", since both parameters should indicate the same point in time. For snow cover melt ("SCD$_{LS}$" and "SCM") this should be the same. However, the relationship is less pronounced. In addition, there is almost always a strong correlation between "Q5%" and "Q10%" as well as "Q90%" and "Q95%".

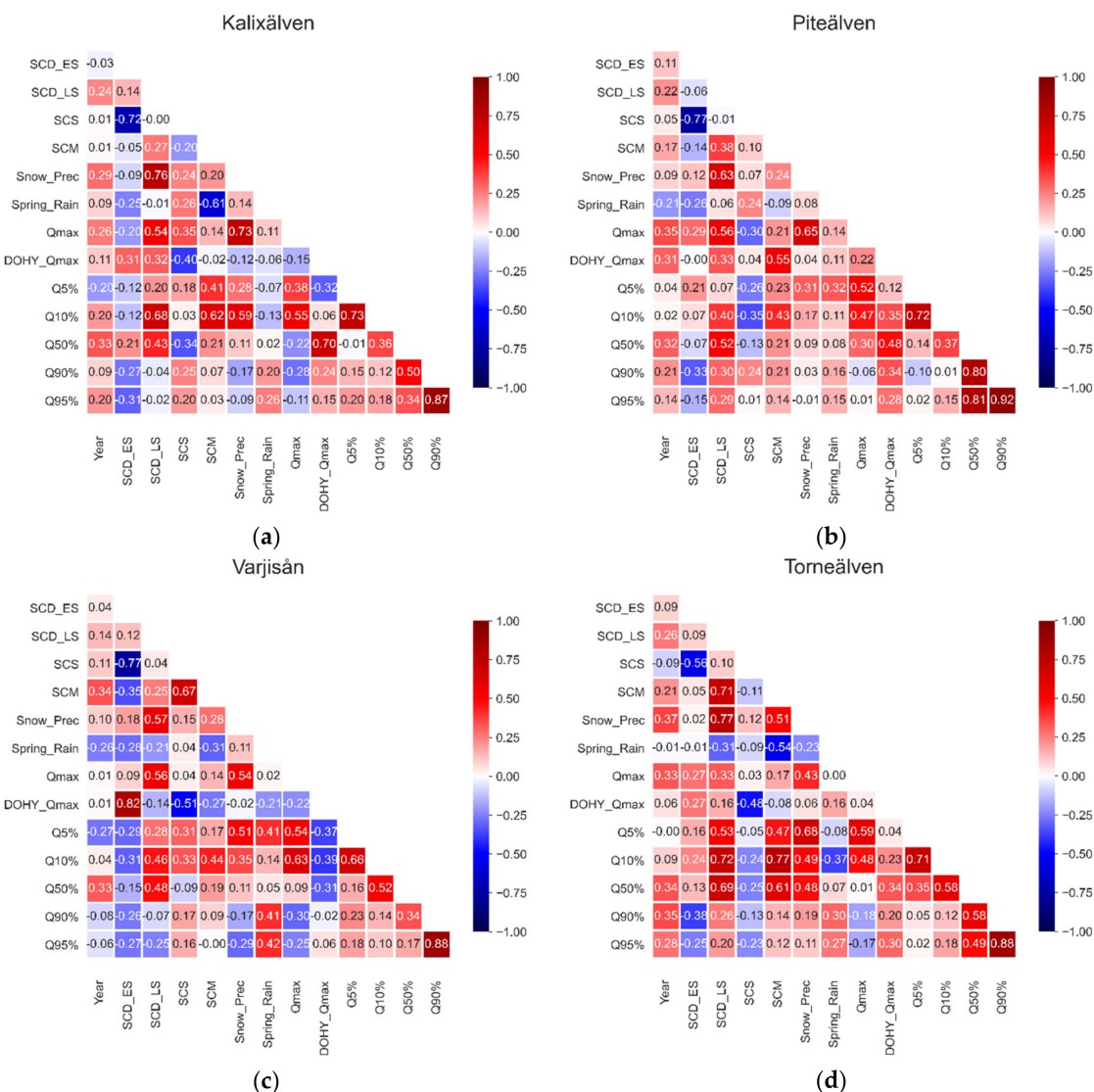

**Figure 12.** Correlation matrices for the timing measures of (**a**) Kalixälven, (**b**) Piteälven, (**c**) Varjisån, and (**d**) Torneälven.

The following conclusions can be drawn from Figure 12 for the first four rivers: the parameter "Year" does not show any clear correlation with other parameters. The two parameters for snow cover start ("$SCD_{ES}$" and "SCS") also have little influence on the spring runoff. Only in the small river Varjisån there is a strong connection between "$SCD_{ES}$" and the time of the peak discharge. There is also a weak relationship between "SCS" and "DOHY_Qmax" at Torneälven. In addition, there are mostly moderate or weak correlations between winter precipitation ("Snow_Prec") and "$SCD_{LS}$" as well as spring precipitation ("Spring_Rain") and "SCM". The most relevant parameter for flood risk, "Qmax", mostly shows only weak or moderate connections to "$SCD_{LS}$" and "Prec_Snow".

The correlation matrices of the remaining four rivers in Figure 13 show similar relationships to those that have already been mentioned. However, there are a few noticeable features. With the exception of the Vindelälven, there is a strong correlation between "$SCD_{LS}$" and "Q50%" in the other three rivers. In addition, in the Ounasjoki there is a strong connection between the time of the discharge peak ("DOHY_Qmax") and "$SCD_{LS}$". The Mounioälven shows at least a weak relationship between these parameters.

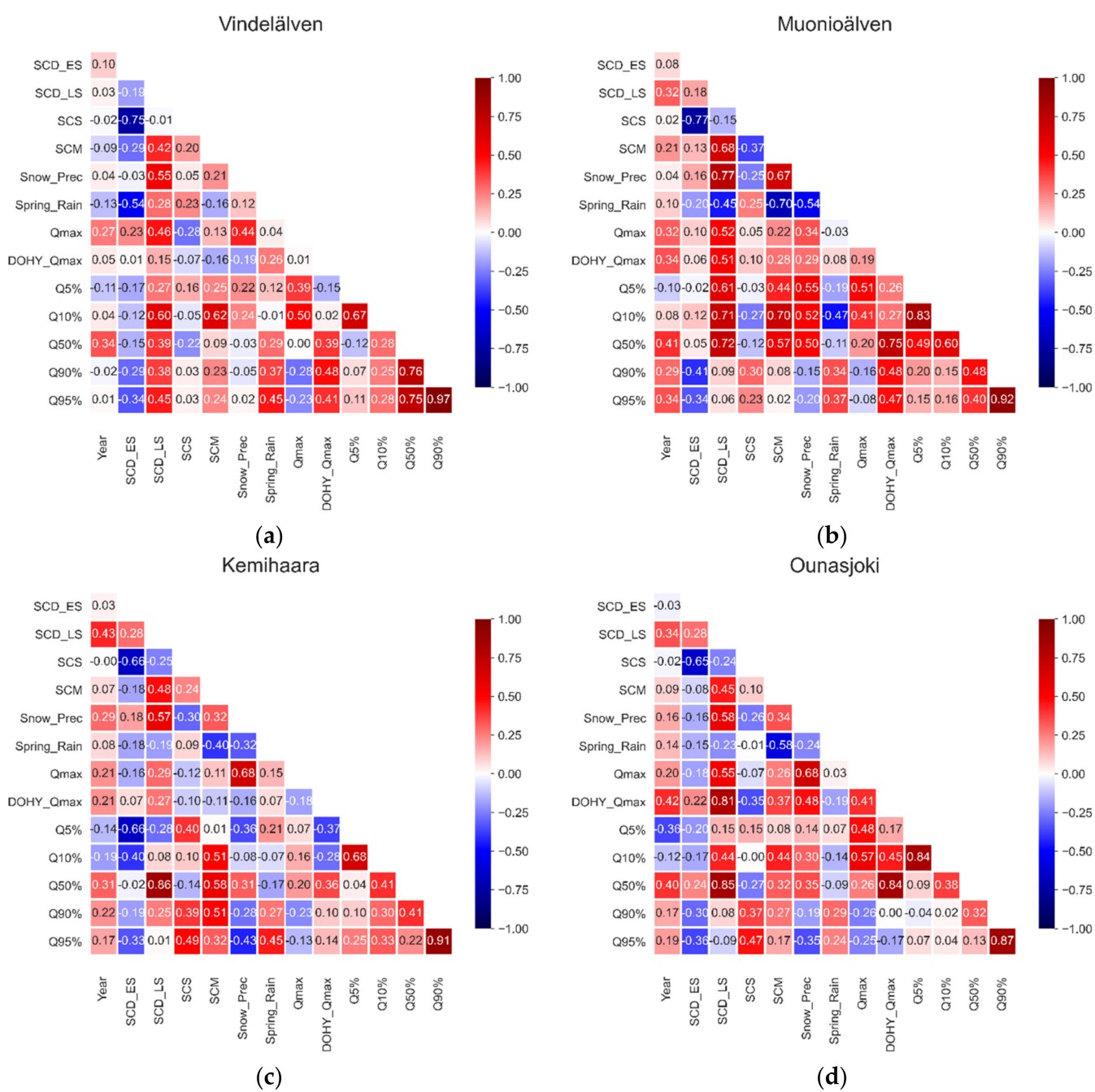

**Figure 13.** Correlation matrices for the timing measures of (**a**) Vindelälven, (**b**) Mounioälven, (**c**) Kemihaara, and (**d**) Ounasjoki.

Table 3 shows the results of the Mann-Kendall tests for the trends of the timing measures in the last 20 years. As was to be expected from the low correlation coefficients between the parameter "year" and the times presented, we observed no trends at the significance level of $p < 0.05$. After increasing the threshold value to $p < 0.1$ (marked with ** in Table 3), we were able to detect trends in three rivers: At the Piteälven, "Qmax" increases by 0.22 m³/s per year. At the Mounioälven, "SCD$_{LS}$" increases by 0.52 days per year, "DOHY_Qmax" increases bay 0.84 days der year and "Q50%" increases by 0.5 days per year. At Ounasjoki, "DOHY_Qmax" shifts by 0.71 days per year. At a significance level of $p < 0.3$ (marked with * in the table), we see especially a trend of "Q50%" for all catchments (between 0.25 and 0.5 days per year).

**Table 3.** Trends resulting from MK tests.

| Parameter | Unit | Kalixälven | Piteälven | Varjisån | Torneälven | Vindelälven | Mounioälven | Kemihaara | Ounasjoki |
|---|---|---|---|---|---|---|---|---|---|
| $SCD_{ES}$ | days/year | 0 | 0.29 | 0.18 | 0.2 | 0 | 0.21 | 0.24 | 0.16 |
| $SCD_{LS}$ | days/year | 0.56 * | 0.24 | 0.45 * | 0.63 * | 0.03 | 0.52 ** | 0.5 * | 0.56 * |
| SCS | days/year | −0.29 | 0 | 0.11 | −0.17 | 0 | −0.1 | 0 | −0.11 |
| SCM | days/year | 0 | 0.25 | 0.5 * | 0.46 | 0 | 0.41 | 0.03 | 0.16 |
| Snow_Prec | mm/year | 1.88 * | 2 | 2.59 | 2.21 * | 1.58 | −0.6 | 0.81 | 0.33 |
| Spring_Rain | mm/year | 1.25 | −0.87 | −1.17 | −0.06 | −0.85 | 1.67 * | 0.36 | 1.94 * |
| Qmax | $(m^2/s)/year$ | 0.09 | 0.22 ** | −0.08 | 0.06 * | 0.07 * | 0.1 * | 0.11 * | 0.06 |
| DOHY_Qmax | days/year | 0.21 | 0.15 | 0.52 * | 0.81 * | 0.33 | 0.84 ** | 0.61 * | 0.71 ** |
| Q5% | days/year | −0.07 | 0.08 | −0.11 | 0 | −0.13 | 0 | 0 | −0.25 * |
| Q10% | days/year | 0.14 | 0 | 0 | 0.13 | −0.03 | 0 | −0.16 | −0.1 |
| Q50% | days/year | 0.31 * | 0.31 * | 0.25 * | 0.43 * | 0.47 * | 0.5 ** | 0.26 * | 0.5 * |
| Q90% | days/year | 0 | 0 | −0.08 | 0.12 * | 0.07 | 0.13 | 0.17 * | 0.07 |
| Q95% | days/year | 0 | 0 | 0 | 0* | 0 | 0.09 * | 0.09 | 0.12 * |

** = *p*-value < 0.1; * = *p*-value < 0.3; no * = *p*-value ≥ 0.3.

### 3.5. Snowmelt Runoff Model

Information on the daily snow cover (divided into 500 m altitude levels), the mean daily air temperatures and the daily precipitation were incorporated into the snowmelt runoff model. The runoff was calculated for each river for each hydrological year. For the sake of clarity; however, only the years with the maximum (red lines) and minimum (green lines) absolute discharge peaks are shown in the following Figures 14–16.

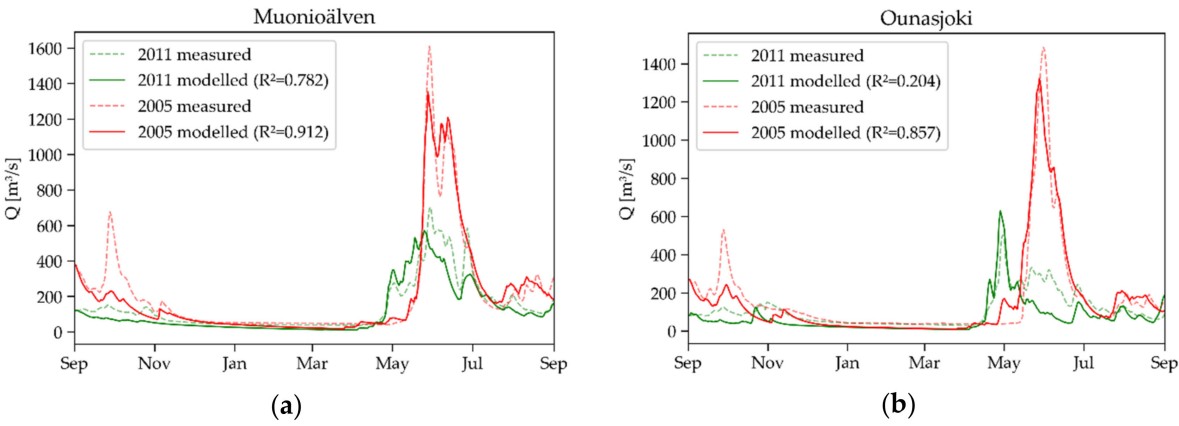

(a)                                                                                     (b)

**Figure 14.** SRM results for the maximum and minimum peak discharge of (**a**) Muonioälven and (**b**) Ounasjoki.

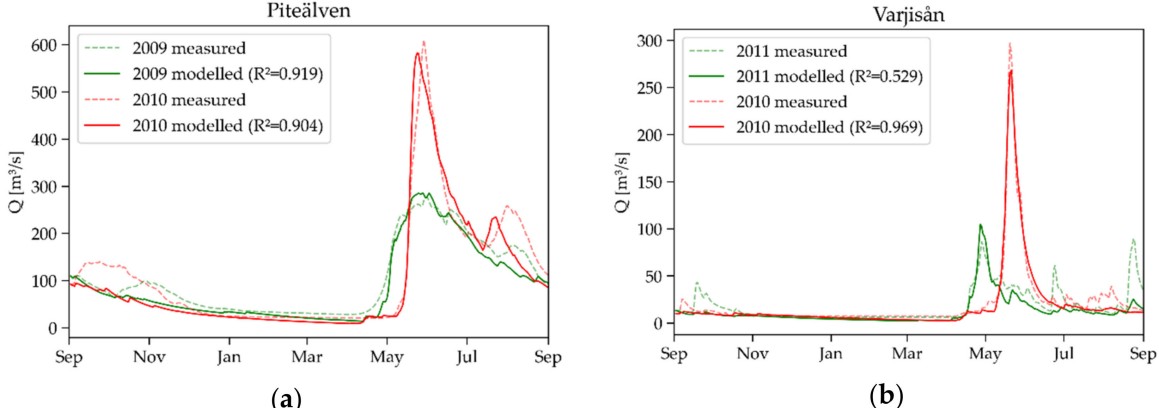

(a)                                                                                     (b)

**Figure 15.** SRM results for the maximum and minimum peak discharge of (**a**) Piteälven and (**b**) Varjisån.

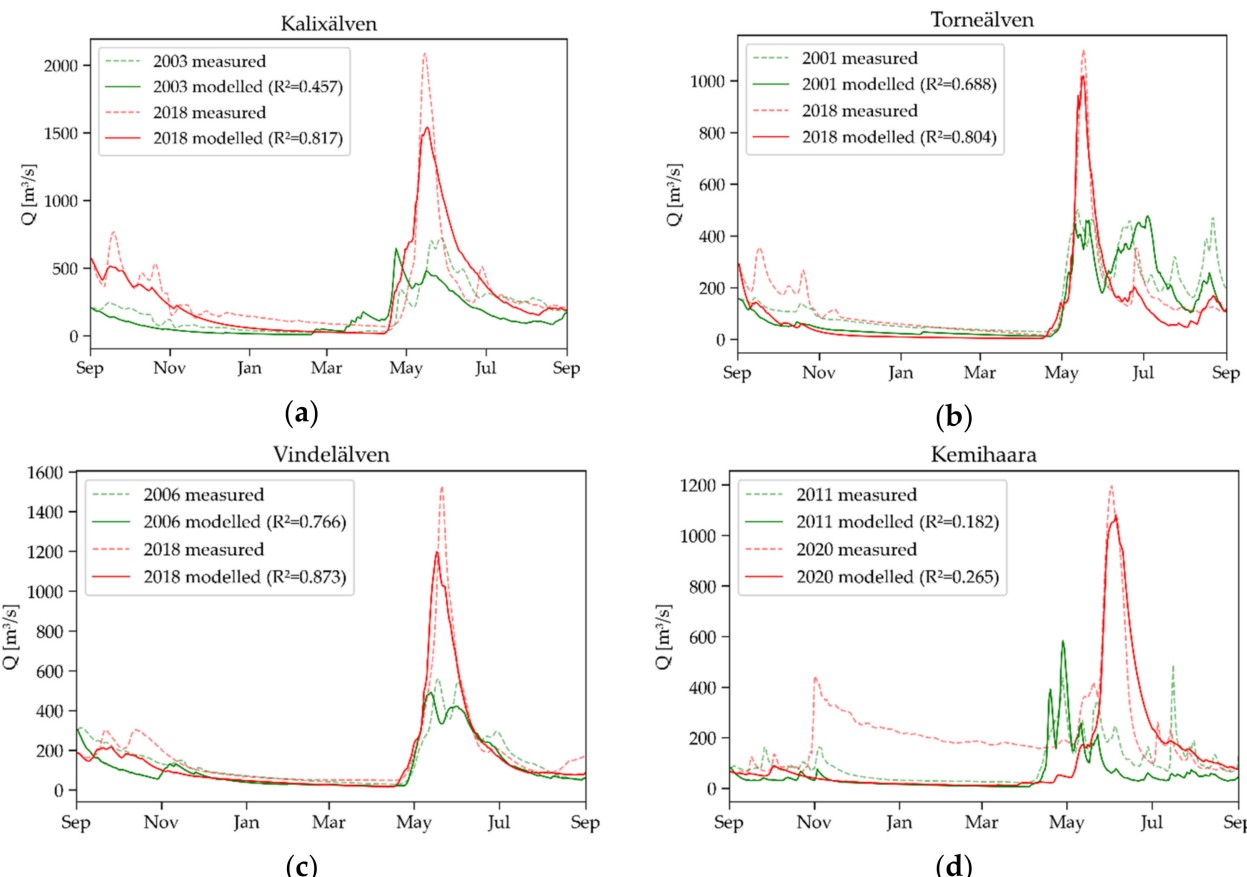

**Figure 16.** SRM results for the maximum and minimum peak discharge of (**a**) Kalixälven, (**b**) Torneälven, (**c**) Vindelälven and (**d**) Kemihaara.

The duration of the spring runoff for 2005 was well determined for both the Mounioälven (Figure 14a) and the Ounasjoki (Figure 14b). The discharge peak was a bit underestimated for both as was the smaller earlier autumn discharge peak. For the minimum discharge in 2011, the spring peak flow itself was better modeled for Ounasjoki, although the coefficient of determination ($R^2$) was very low.

The modeling results of the spring flood of Piteälven (Figure 15a) and its tributary Varjisån (Figure 15b) for the maximum peak flow in 2010 showed overall a good agreement between modeled and measured runoff. The absolute peak runoff was again a bit underestimated and the time of the peak was estimated a few days too early for Piteälven. For Piteälven and Torneälven discharge for the first ~90 days was underestimated. The minimum discharge peaks in 2009 (Piteälven) and 2011 (Varjisån) could be depicted with sufficient accuracy.

For the major flood year of 2018, the model performed well for the rivers Torneälven (Figure 16b) and Vindelälven (Figure 16c). For Kalixälven (Figure 16a) and Kemihaara (Figure 16d), the model result could be improved. The peak flow of Kalixälven was underestimated by ca. 500 m$^3$/s and its onset was estimated too early. For Kemihaara, the location of the spring runoff peak was correctly estimated, but the model could not handle the large amount of base runoff in winter.

## 4. Discussion

Since in this study we defined a fixed point in time for the differentiation between Early (SCD$_{ES}$) and Later Season Snow Cover Duration (SCD$_{LS}$) for each catchment area, certain abstractions [48] had to be made. The catchment areas are therefore viewed as a whole, regardless of the topography. This is particularly critical for high relief energies such

as the catchment area of Kalixälven, in which Kebnekaise, Sweden's highest mountain, lies. Both $SCD_{ES}$ and $SCD_{LS}$ will differ over a large range within the different elevations. The abstractions can be seen when we compare the timing of $SCD_{ES}$ or $SCD_{LS}$ derived from GSP and SCS or SCM from the air temperature developments. Regarding the satellite-derived snow cover duration using GSP, it turned out to be useful to divide the snow cover duration into an "early" and a "late" part, separated by the date of maximum snow coverage [29]. The $SCD_{ES}$ is mostly affected by the polar night; hence this information is less accurate. The values from GSP are mostly several days behind SCS and SCM derived from air temperature. An explanation would be that the snow accumulation (ablation) first needs negative (positive) air temperatures to occur and the time until full coverage (melt) depends on the precipitation amount (snow pack). A major issue in these northern regions is the long period of time over which interpolation has to be carried out (due to cloud cover or polar night). The further away the days are to a snow-free pixel, the less its reliability becomes. In the latest version of Global SnowPack, an accuracy layer is therefore created that is based on these "days to cloud-free" and based on the topography.

The meteorological parameter solid precipitation ("Snow_Prec") correlates much better with the $SCD_{ES}$ derived from GSP, since it is directly linked to the disappearance of the snow cover. We also noticed that the runoff measures "Q90%" and "Q95%" have no relationship with snow melt, which has also been proven by [1]. Our most important measure; however, were "Qmax" and its timing "DOHY_Qmax" [49]. The main idea of the study was to determine whether extreme events can only be detected with the snow cover alone. Since we mostly found only weak relationships (correlation coefficient between 0.4 and 0.6) between "Qmax" and GSP-derived $SCD_{LS}$, further information is required.

The reasons for hydrological extremes (exceptionally high and low runoff) can therefore be found in the combined consideration of the hydrometeorology and the satellite-derived snow situation [50,51]. Within the period under consideration we want to look at the causes of the lowest and highest spring floods. Blahušiaková et al. [43] identified in Central Europe two reasons for snow drought: 1) average snow precipitation and unusually high air temperatures leads to below-average snow accumulation or 2) low or normal air temperature and low winter precipitation. The latter was the reason for all minimal outflows of the rivers considered in this study. With regard to the occurrence of the highest discharge values in the period under review, we could distinguish between four types associated with the four flood years (Table 4).

**Table 4.** Development of measures for different flood types during record floods.

| Type | Year | Rivers | $SCD_{ES}$/SCS | $SCD_{LS}$/SCM | Snow Precipitation | Remarks |
|---|---|---|---|---|---|---|
| 1 | 2005 | Muonioälven, Ounasjoki | Normal | 2 weeks later | 35% higher | - |
| 2 | 2010 | Piteälven, Varjisån | 3 weeks earlier | Normal | 20% higher | - |
| 3 | 2018 | Kalixälven, Torneälven, Vindelälven | 1 week earlier | Normal | Normal (up to 10% higher) | A drastic increase in air temperature led to very rapid melting |
| 4 | 2020 | Kemihaara | 2 weeks earlier | 3 weeks later | 60% higher | - |

As also observed by Shi et al. [1], air temperatures fluctuations (especially in spring) are the major causes for severe flood events (like in 2018), since they define the onset and end of snow accumulation and hence also snow storage. This leads us to the risk of rain-on-snow (ROS) events [25,49,52], which can occur more frequently if the snow cover lasts well into spring. These events can also have drastic effects on e.g., reindeers, as solid ice layers form at the bottom of the snow cover, so-called "basal ice", and prevent them from grazing [53]. The occurrence of basal ice has also a significant impact on hydrological modeling since melt water cannot infiltrate. In the examples presented, a ROS event could

have taken place at Kalixälven in 2018. The maximum runoff here was 40% higher than modeled and the duration of the flood runoff was greatly reduced. The assessment of these events requires extensive fieldwork, which was not possible within the scope of this investigation. Accordingly, further investigations in this area are recognized as a knowledge gap [54].

The Trend analysis revealed an increase of "Qmax" for Piteälven, an increase of "DOHY_Qmax" for Muonioälven and Ounasjoki, and an increase of "SCD$_{LS}$" and "Q50%" for Muonioälven. The shift of runoff peaks had been observed by Dankers and Christensen [55], who were investigating the Tana River (adjacent basin north of the Kemihaara). However, they and Trenberth [56] also recognized a general shortening of the snow season, where the opposite was the case in our study period. Although the trends for winter precipitation were not significant, all point to an increase in winter precipitation of up to 2mm per year, which corresponds to the general precipitation trend in Northern Europe [56]. Several studies examined the relationship between snow cover, hydrometeorology and runoff using Mann-Kendall tests [18,43,44]. In high and moderate latitudes of the northern [43] and southern [44] hemisphere, it was found that (winter) precipitation increases as does the air temperature and thus also the runoff. In the Western Himalayas; however, Atif et al. [18] the opposite observed, which in turn is due to local climatic conditions.

The rather simple snowmelt runoff model from Martinec at al. [34] was generally well suited to depict the measured runoff values using only snow cover information from Global SnowPack [31] and hydro-meteorological station data. Most important was the correct representation of the spring floods (peak, volume, and duration). When simulating the snow-free runoff there are certainly better models that incorporate surface properties and soil information [57]. The SRM is widely used e.g., in the Himalaya [58,59], in India [60] and in Morocco [47]. The coefficients of determination were in general higher for the maxima of the peak values and lower for the minima. Only for the river Kemihaara we achieved poor results for both. The extraordinarily high basic runoff before snow melt turned out to be a measurement error (according to SYKE). After snow melt onset, the discharge measurements are correct again. Regarding the general underestimation of the maximum discharge, Dankers and Christensen [55] indicates that the contribution of spring precipitation is underestimated in these models. This would be an explanation for the poor model results of the minimum peak discharge of Ounasjoki in 2011, where only 59% of the mean snow precipitation fell, but the spring precipitation was 42% above the average.

## 5. Conclusions

In this study, we presented how satellite-derived snow cover information can be linked to the occurrence of hydrological extreme events. The study was carried out on eight unregulated river catchments in northern Fenno-Scandia (Sápmi), which received a large amount of snow in the winter of 2019/2020. There is a high variability of spring snow cover in the eight catchments studied, with a greater variability in the Finnish catchments (Figure 6). However, the discharge data show greater variability in the timing and longevity of the spring flood in the Swedish catchments draining the mountains (with the notable exception of the small Vajisån catchment) and Muonioälven in Finland: the most mountainous and westerly of the Finnish catchments. Statistical analysis of the data showed that remote sensing-based measures of snow cover and snow melt were reasonably well correlated with discharge measures, especially those at the onset of spring flood discharge. Peak discharge ("QMAX") was often best correlated with the amount of snowfall in the season ("Snow_Prec"). The relationship between "Spring_Rain" and the timing of "Q95%" discharge indicates rain-on-snow events are import later in the season. This can also explain why the discharge modeling worked well, but sometimes could not resolve extreme events or discharge spikes.

Our results show that remote sensing data are important complements to meteorological observations and can help in the explanation of discharge time series in Arctic and sub-Arctic catchments. We have shown that meteorological data alone cannot explain

spatio-temporal patterns in discharge data nor can modeling using meteorological data explain extreme events. This argues for combined approaches that include geospatial data on snowpack coverage or snow depth for any comprehensive investigation of discharge during the spring flood. Particularly in the early snow season, there are great uncertainties due to the interpolation of the polar night. In the future, in addition to the optical satellite data, coarse-resolution data from microwave sensors will also be used for snow detection.

The trend analysis only showed a significant increase in the spring flood peak at the river Piteälven and a general delay in the spring flood for the river Mounioälven. A better result can only be achieved with a longer time series. The snow cover could be obtained from the time series of AVHRR data, which is currently being reprocessed for whole Europe within the TIMELINE project at DLR [61] and will be made available to the public.

With the simple Snowmelt Runoff Model, good results were achieved, especially when depicting floods. This model shows that the use of publicly freely available data sets enables relatively accurate modeling of extreme snow-related water events. Since the occurrence of Rain-on-Snow (ROS) events worsens the modeled results, further investigations are planned here.

**Author Contributions:** S.R. performed the MODIS data processing with GSP, the data analysis and statistical interpretation and wrote the majority of the manuscript. M.S.W. developed a Python-based variant of the Snowmelt Runoff model and was responsible for the description, implementation and interpretation of the modeling results. J.I. provided the data for the Finnish catchment areas, helped with the selection of suitable test waters and with the interpretation of the results for the manuscript. I.A.B. compiled all available data for the Swedish rivers and made an important contribution to the identification of suitable test areas, the knowledge of local conditions and the correction of the manuscript. A.J.D. provided help with the study design and with the compilation and correction of the manuscript, and he also established contact with the co-authors involved. All authors have read and agreed to the published version of the manuscript.

**Funding:** This research has been funded by the DLR's "Polar Monitor" project. I.A.B. was funded by the Swedish National Space Agency.

**Institutional Review Board Statement:** Not applicable.

**Informed Consent Statement:** Not applicable.

**Data Availability Statement:** Data on runoff, meteorological parameters and the MODIS snow information are freely available from the sources mentioned in the text. The Snowmelt Runoff Model used can be downloaded from https://data.nal.usda.gov/dataset/snowmelt-runoff-model-windows-winsrm. The results of the latest version of the DLR product Global SnowPack will soon be available online as part of the "Polar Monitor" project.

**Acknowledgments:** The authors would like to thank the editors and three anonymous reviewers who have increased the quality of the manuscript with their suggestions for improvement.

**Conflicts of Interest:** The authors declare no conflict of interests.

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
