# Peer review of "Remote Sensing of Snow Cover Variability and Its Influence on the Runoff of Sápmi’s Rivers"

_geosciences, doi:10.3390/geosciences11030130_

Round 1
Reviewer 1 Report
This is potentially an interesting study, and the authors have assembled a valuable dataset, but I find the focus of the work somewhat elusive. I think the introduction needs a bit of rewrite to clarify the aims so that the reader is not left wondering why some part of the analysis has been performed. The results are somewhat overinterpreted in view of the weak correlations found, and this section should be reduced.
Use of the MODIS daily snow cover products (M*D10A1) makes sense. But wouldn’t the 8-day products be preferable (M*D10A2) since they have fewer cloud-cover issues and most of the analysis has as weekly rather than daily resolution? Why not discussed?
Interpolation of missing data because of cloud (and presumably also polar night, though this isn’t explicitly discussed) introduces some potential error into the binary snow/nonsnow classification. Lacks discussion.
SCS and SCM are defined by equations 3.1-3.2 in a way that relies on snow cover remaining throughout the intervening period. Can this be safely relied on?
SCS and SCM are defined in lines 174-178 in terms of the cumulative temperature. This could be a hypothesis to be tested, that the temperature-derived dates described here will correspond to SCS and SCM, but it seems odd to actually define them this way. (See also comment on lines 411-414.)
Detailed comments and queries (by line number)
(38) ‘will have to cope with’ - the verb needs an object.
(52) ‘studying’ → ‘application to study of’ or similar
(59) ‘flood protection and hydropower’ or ‘flood protection or hydropower’? I mean, are the two necessarily combined?
(93) ‘Amount of days’ → ‘Number of days’
(97) ‘a NDSI’ → ‘an NDSI’
(98) Maybe explain the basis of these empirical thresholds, and how their parameters were determined.
(Eqq 1.1 & 1.2) Please use multiplication symbols not asterisks
(101) ‘by merging both grids’. I don’t understand this – do you mean MOD and MYD products for the same day? What is the merging logic if one says ‘snow’ and the other ‘non-snow’?
(106-111) Is this elevation-dependent method of interpolating snow cover applied the same way across the whole tile? One might expect the threshold elevations to depend on latitude and distance from the sea.
(112) ‘no more cloud pixels’. Not clear whether this processing also accounts for ‘missing data’ pixels.
(121) ‘due to maximum snow coverage’ - how was this established?
(132) What is the colour bar in fig 2? The colours look like topography but the numerical values can’t be metres.
(146) What are the numbers in the ‘basin’ column of table 1?
(157-159) Odd to use Q as a symbol to represent a date when it is also used for discharge rate.
(159) ‘accordingly’ → ‘similarly’.
(169) ‘analyzes’ → ‘analyses’
(214) ‘evaluated in the ranges from 0.05 to 0.99’ - presumably values can be expected to lie within this range?
(214) ‘ranges’ → ‘range’
(214) ‘for the combination’ → ‘to find the combination’
(228) ‘part’ → ‘parts’
(231) ‘particular’ → ‘particularly’
(234) Fig 5: basin delineations are too hard to see. Can the line colour be changed?
(239) comma needed after ‘3.2’
(240) ‘values stagnate’. Isn’t this concerning and potentially distorting results? Could dates with higher uncertainty for % SCA be flagged somehow?
(249) Fig 6 (and later similar figs). Consider whether the dates on the vertical axis should run bottom to top rather than top to bottom?
(251) Fig 6 caption needs also to explain what the solid and dashed lines represent (it’s in the main text but the figure should be understandable without reference to that).
(299) ‘due to low winter precipitation...’ it’s probably true, but this is interpretation and needs to be somewhat separated from the presentation of the results.
(301) ‘led to rapid melting...’ - again, interpretation.
(303) Fig 10. The lines on the precipitation diagrams are hard to see around SCS. Maybe change the colour of both sets of lines on this diagram (e.g. to white)?
(304) Explain the meaning of the lines in the fig caption.
(305-314) ‘is attributable to… had the same reason… is due to a larger amoun… due to a rather large amount...’ Again, these are interpretation and may require some justification. The para beginning in line (319) is more cautiously worded and a better model for this I think.
(318) Explain the meaning of the lines in the fig caption.
(332) ‘normally are’ → ‘normal, are’
(324) ‘<10’ I’m not generally in favour of using mathematical notation in text and think this would be easier to read as ‘less than 10’ but I suppose that is a question of house style.
(334-336) I am not sure how useful this categorisation of regression coefficients is.
(356) ‘relationships that have’ → ‘relationships to those that have’
(363) ‘timers’ → ‘times’ ?
(361-370) I would need to be convinced that this much detail is worth presenting since, as the authors note, the regression relationships are not significant at the conventional 0.05 level. I certainly don’t think we can read much into a relationship that only has p<0.3 significance.
(363) why is this unfortunate?
(384) and subsequently. It might be helpful to show months on the axes of the figure, rather than DOHY.
(406) ‘there for’ → ‘therefore’
(411-414) This discussion of the relationship between the temperature-derived and satellite-derived snow season dates reads somewhat oddly unless there had been an initial, preferably justified, hypothesis that the two kinds of definition would in fact coincide. The results are described here do not seem particularly surprising , for the reasons suggested.
(419-420) ‘The main idea of the study...’ I don’t think this is clear from the introduction.
(423) ‘reasons in’ → ‘reasons for’
(460) ‘correct mapping’ - is ‘mapping’ the right word here?
Reviewer 2 Report
The Authors rightly note that snowfall prediction and spatial distribution of snow cover is a complex problem, aspecially in the light of climate changes. It is stressed that currently snowmelt is the main mechanism of flood generation in northern Fenno-Scandia, the study area. Based on selected events, four sinoptic types concerned with flood were identified and hydrometeorological conditions triggering the process were simulated in regional scale of eight selected catchments. The Authors used well known snowmelt runoff model SRM for the simulations.
Reviewer 3 Report
Authors in the manuscript "Remote sensing of snow cover variability and its influence on the runoff of Sapmi's rivers" used different scientific methods to understand the role of the snow cover in the hydrology of the chosen catchments. Authors chosen catchments located in Finnish Lapland and northern Sweden. To examine the temporal development of the snow cover and runoff, the authors investigated the temporal development of the various robust time measures. The analysis was done based on the data from 2001 to 2020. These data were used to feed the simple snow runoff model.
The manuscript is very interesting. Comparison of the different data and methods to understand the relationship between snow accumulation and runoff (especially extreme runoff) can be important for the local authorities and other scientists working in different regions.
The manuscript is well written but has to be improved.
Abstract and Introduction: please clearly indicate the aim of this research. What is the hypothesis if any?
Line 54: please add "space" between coma and "North".
Line 85 and 86: change ordinal numeral abbreviation ("st") to uppercase.
Maps: add North arrows
Table 1: Column "Basin" - what are these numbers? If those numbers are the official number of each basin, you have to mention this. In my opinion, it is worth to add the country for each basin and sort this catchment with elevation.
The column description "Elevation [m MSL]": what does it mean "MSL". In my opinion should be "asl".
Line 161 and in the whole text authors writing about the "temperature". I guess that is about the air temperature and not about the ground temperature or any other temperature. Please add in whole text "air" when you are talking about the air temperature.
Line 163: remove the space between "for" and " Sweden"
Line 172: Please write why you are using the value of 0.65 K /100 m? Is this vertical gradient calculated based on the real data? This information has to be given in the text.
Catchments names in the text and in the figures. In my opinion, the "Basin number", which is given together with names of the catchments is not necessary. I suggest removing this from the text. The number is mentioned in Table 1. Using number and the name makes chaos.
Line 260: "X" sign should be like as is at the figures - "x".
Line 273 to 280: Remove the basin numbers and move the figure numbers to the end of the brackets.
Line 320: What does it mean "surface temperature"? Is this snow surface temperature?
Why authors don't take into consideration wind speed and direction? A strong wind (especially warm wind) affects the melting.
Table 3: Please change the basin numbers to the basin names. If I good understand the number is given without asterisk then the p-value is 0.05. I suggest adding under the table that "no * - p-value = 0.05".
Line 438: The authors mention rain-on-snow events. This ROS can happen during the winter thaw periods. Then rain creates ice layers between the snow layers and between ground surface and snow layer. This basal ice can change the runoff conditions of the water from melting snow and rain events. Did the authors measure or observed basal ice? Can you describe how this ROS and basal ice can influence the modelled results?
Please add Conclusions chapter.
Round 2
Reviewer 1 Report
The authors have done a good job in addressing the concerns I had with the first version of this manuscript. The aims have been clarified. It's a useful piece of work.